# New Insights into Lichenization in Agaricomycetes Based on an Unusual New Basidiolichen Species of *Omphalina s.* str.

**DOI:** 10.3390/jof8101033

**Published:** 2022-09-29

**Authors:** Tingting Zhang, Xinyu Zhu, Alfredo Vizzini, Biting Li, Zhenghua Cao, Wenqing Guo, Sha Qi, Xinli Wei, Ruilin Zhao

**Affiliations:** 1State Key Laboratory of Mycology, Institute of Microbiology, Chinese Academy of Sciences, Beijing 100101, China; 2College of Life Sciences, University of Chinese Academy of Sciences, Beijing 100049, China; 3Center of Excellence in Fungal Research, Mae Fah Luang University, Chiang Rai 57100, Thailand; 4Dipartimento di Scienze Della Vita e Biologia Dei Sistemi, Università di Torino, Viale P.A. Mattioli 25, 10125 Torino, Italy; 5College of Life Science and Technology, Xinjiang University, Urumqi 830000, China; 6The Wanan County Second Middle School, Wanan 343800, China

**Keywords:** agaricales, basidiolichen, basidiomycota, fruiting body, green algae, phenotype, systematics, new taxon

## Abstract

The genus *Omphalina* is an ideal genus for studying the evolutionary mechanism of lichenization. Based on molecular phylogeny using ITS and nuLSU sequences by means of Bayesian and maximum likelihood analyses and morphological examination, combining the existence of green algae in basidiomata stipe and a *Botrydina*-type vegetative thallus, we described a bryophilous new basidiolichen species, *Omphalina licheniformis*, from a residential area of Jiangxi Province, China. This finding of unusual new basidiolichen species updated our understanding of the delimitation of *Omphalina*, indicating that both non-lichen-forming and lichen-forming fungal species are included simultaneously. The presence of algal cells in the basidiomata should receive more attention, as this would be helpful to distinguish more potential basidiolichens and explore the cryptic species diversity. This work provides new insights and evidence for understanding the significance of lichenization during the evolution of Agaricomycetes.

## 1. Introduction

Lichens are symbionts of fungi (mycobionts) and algae and/or cyanobacteria (photobionts), among which only 0.9% species belong to the Basidiomycota [1]. *Omphalina* Quél. is an undisputedly important genus when talking about basidiolichen species, because it included both non-lichenized and lichenized species originally, and was subsequently separated into non-lichenized *Omphalina s.* str. and lichenized genera such as *Lichenomphalia* Redhead, Lutzoni, Moncalvo & Vilgalys [2] and *Agonimia* Zahlbr. (syn. *Marchandiomphalina* Diederich, Manfr. Binder & Lawrey [3]). In addition, *Omphalina* also includes saprophytic, parasitic, and bryophilous species [4]. Therefore, *Omphalina* has been regarded as an ideal genus for studying the evolutionary mechanisms associated with lichenization [4,5].

Recently, molecular phylogenetic analyses pointed out that the classical concept of *Omphalina*, mainly based on morphological features [6,7,8,9], includes several omphalinoid genera nested inside the order Agaricales [2,10,11,12,13,14], as well as Hymenochaetales Oberw. [15,16,17,18]. Inside the Agaricales there are omphalinoid taxa in the suborders Hygrophorineae (family Hygrophoraceae, subfamily Lichenomphaloideae [13]), Marasmiineae (family Porotheleaceae [19,20]) and Tricholomatineae (family Omphalinaceae [21]). Rickenellaceae is the family encompassing omphalinoid taxa in the Hymenochaetales [17,18,22]. *Omphalina* was restricted to the species phylogenetically related to *O. pyxidata* (the conserved lectotype of *Omphalina* [2,10,23,24]), which typically show reddish brown, rusty, or orange-brown tinges on the pileus and stipe, and a non-concolorous hymenophore [2]. Micromorphological features, such as non-amyloid spores, sub-regular to irregular hymenophoral trama and pileipellis with encrusting pigment, are shared by all members of the genus *Omphalina* s. stricto [2,9,25,26,27,28,29]. This genus is sister to *Infundibulicybe* [14,21,30,31,32], and together they form the family Omphalinaceae [33] in the suborder Tricholomatineae.

Basidiolichens are mainly distributed in five orders of Agaricomycetes, viz. Agaricales, Atheliales, Lepidostromatales, Cantharellales and Corticiales, among which Agaricales and the family Hygrophoraceae within this order accommodate most of the basidiolichen species [1]. In China, thirteen basidiolichen species, including five *Dictyonema* spp. (Hygrophoraceae, Agaricales), four *Lichenomphalia* spp. (Hygrophoraceae, Agaricales), and four *Sulzbacheromyces* spp. (Lepidostromataceae, Lepidostromatales), have been discovered [34,35,36].

A cluster of bryophilous *Omphalina* basidiomata was found in a residential area of Jiangxi Province, China. The phylogenetic analyses of nrDNA ITS and nuLSU sequences also confirmed it to be an unknown *Omphalina* species. However, interestingly, it was found in the stipe existence of green algae; moreover, far fewer vegetative thalli consisting of green tiny globules (*Botrydina*-type) were also seen near to the hairs at the base of stipe. Therefore, an unusual new basidiolichen species of *Omphalina* is described and reported here. This finding indicates that *Omphalina* s. str. still consists of both non-lichenized and lichenized species with very a close phylogenetic relationship, and further provides new insights into the lichenization in Agaricomycetes.

## 2. Materials and Methods

### 2.1. Taxon Sampling and Morphological Examination

Five basidiomata specimens were collected from a residential area of Wan’an County, Jiangxi Province of China (Figure 1), and are preserved in the Herbarium Mycologicum Academiae Sinicae, Beijing, China (HMAS). Morphology and anatomy were examined using a MOTIC SMZ-168 stereomicroscope and a LEICA M125 dissecting microscope equipped with a Leica DFC450 camera.

Photographs of fresh specimens were taken immediately in the residential area, and the basidiomes were gathered. The morphological characteristics, including cap, stipe, pileus, and odor, were recorded. Specimens were dried in an electrical food drier at 55 °C to ensure that no moisture was left, and then were sealed in plastic bags.

To observe anatomic characteristics, parts of dried specimens were cut and mounted in 5% KOH and stained with 1% Congo Red. Anatomic characteristics including basidiospores, basidia, cystidia and pileipellis were observed under an Olympus CX31RTSF microscope (Made in Philippines, Tokyo, Japan) with at least 20 records. Data were analyzed and recorded as X = the mean of length by width ± SD, Q = the quotient of basidiospore length to width, and Qm = the mean of Q values ± SD. All of the protocols of the morphological study followed Largent’s methodology [37].

### 2.2. DNA Extraction, PCR, and Sequencing

DNA was extracted from two fresh basidiomata (Appendix A) by means of the modified CTAB method [38]. PCR was performed to amplify two gene loci: nuclear ribosomal DNA internal transcribed spacer (ITS) and large subunit (nuLSU), using primers ITS4 and ITS5 [39], and LR0R and LR5 [40], respectively. The PCR procedure followed Yang et al. [41].

### 2.3. Sequence Alignment and Phylogenetic Analysis

A total of 136 DNA sequences including four new sequences were used in this study (Appendix A). Representative species of the lichenized genera of Hygrophoraceae (Agaricales), non-lichenized genera of Agaricales, and other related orders in the Agaricomycetes were chosen in the phylogenetic analyses. *Multiclavula* spp. (Clavulinaceae, Cantharellales) were taken as the outgroup.

Raw sequences were firstly assembled and edited with SeqMan [42], and then aligned using MAFFT v.7 [43]. We used the program Gblocks v.0.19b [44,45] to remove ambiguously aligned sites. The congruence of the two loci (ITS and nuLSU) was tested as described previously [46,47]. All maximum likelihood (ML) and Bayesian inference (BI) analyses were performed using the GTR + I + G model selected by jModelTest 2 [48]. The ML analysis involved 1000 pseudoreplicates with RAxML v.8.2.6 [49]. The BI analysis was performed using MrBayes v. 3.2.7 [50,51] with two parallel Markov chain Monte Carlo (MCMC), each using 5 million generations and sampling every 1000th generation. We used TRACER v.1.7.2 [52] to examine the standard deviation of split frequencies less than 0.01, reflecting the fact that the two trees differed very little, and the parameters converged. The 50% majority rule consensus tree was generated after discarding the first 25% as burn-in.

Phylogenetic analysis was run on the Cipres Science Gateway (http://www.phylo.org, (accessed on 18 July 2011)) and visualized using FigTree v.1.4.3 (http://tree.bio.ed.ac.uk/software/figtree, (accessed on 28 August 2014)). The clades with bootstrap (BP) values above 75% or posterior probability (PP) values above 0.95 were considered highly supportive.

## 3. Results

### 3.1. Phylogenetic Analysis

The aligned matrix contained 2011 unambiguous nucleotide (1129 ITS and 882 nuLSU) position characteristics for the full dataset of 93 members. BI and ML phylogenetic trees were constructed, and they had similar topological structures. The RAxML tree is shown in Figure 2 with both bootstrap support (BS) and posterior probability (PP) values of BI analysis. In the tree, all of the *Omphalina* species clustered into a well-supported monophyletic clade (BS 100/PP 1.00), obviously separated from other groups, in which the two samples (Coll. Nos. JX001 and ZRL20220005) are included and co-formed into a separate branch (BS 100/PP 0.99), indicating that this is a new species also supported by the morphological characteristics (see below). The BI phylogenetic tree and two single-gene-locus RAxML trees are shown in Appendix A.

### 3.2. Taxonomy

The genus *Omphalina* is known to be a non-lichenized basidiolichen genus, because the original lichenized species contained in this genus have been separated out and formed totally different other genera such as *Lichenomphalia* [2]. However, an *Omphalina* new species was found to be lichenized in this study, and is described below. Therefore, the definition of the genus *Omphalina* also needs to be redefined as the genus with lichenized species in some cases.

*Omphalina* Quél., Enchir. fung. (Paris): 42 (1886)

Type species: *O. pyxidata* (Bull.) Que’l., *incertae sedis*, Agaricales, Agaricomycetes, worldwide. This genus is characterized by small basidiomes, pileus convex to umbilicate, reddish brown tinged, smooth, without scales; lamellae decurrent, paler and well-developed; stipe central, reddish brown tinged; hymenial and pileal cystidia absent or sparse, and presence of clamp-connections [2,53,54]. Sometimes lichenized.

*Omphalina licheniformis* X.L. Wei, Z.H. Cao & R.L. Zhao, **sp. nov.** (Figure 3 and Figure 4)

Fungal Names No.: FN571066

Etymology: The epithet ‘*licheniformis*’ indicates that this species is a lichen-forming fungus.

Typus: China. Jiangxi Province, Ji’an City, Wan’an County, 26.465265° N 114.798753° E, 65 m alt., on moss *Hyophila involuta*, 10 March 2022, Z.H. Cao & W.Q. Guo JX001 (HMAS–L 154705, holotype), ZRL20220005 (HMAS 281952, isotype).

Diagnosis: *Omphalina licheniformis* is distinguished from other species of this genus by having smaller basidiospores (4.9–5.5 × 3.8–4.6 μm) and presenting distinctive cheilocystidia. It is also characterized by the presence of far fewer vegetative thalli consisting of green tiny globules (*Botrydina*-type) near to the hairs at the base of the stipe, and green algae in the stipe.

Description: Thalli not obvious, *Botrydina*-type, globular, with globules clustered, very few and tiny, plump when wet, yellow-green to green, c. 80 µm in diam.; globules consist of unicellular green alga enveloped by the hyaline hyphae, hyphae 2.5–5 µm wide, algal cells 6–10 µm in diam.

*Basidiomes* small, omphalinoid. *Pileus* 7–15 mm in diam., subhemispherical when young and subfunnel when mature, distinctly depressed, hygrophanous, red brown to yellowish brown, edge paler, becoming alutaceous when drying, margin involute when young, wavy and striate when mature. *Context* thin, up to 1 mm, concolorous with pileus surface. *Lamellae* decurrent, distant, thick, forked and anastomosing, concolorous with pileus, but paler. *Stipe* 15–20 × 3–5 mm, cylindrical, hollow, smooth to white fibrillose, concolorous with the pileus, darker at the middle and lower parts with white mycelium-like cilia. *Smell* and *taste* indistinct.

*Basidiospores* 4.9–5.5 × 3.8–4.6 μm, [x = 5.2 ± 0.2 × 4.1 ± 0.2, Q = 1.2–1.3, Q_m_ = 1.3, n = 20], broadly ellipsoid to ellipsoid, rough with granular contents, smooth, thick-walled, hyaline. *Basidia* 25.0–27.4 × 3.3–4.7 μm, clavate, hyaline, (2-)4-spored, smooth. *Cheilocystidia* 27.1–35.7 × 3.7–7.0 μm, cylindrical and flexuose, narrowly clavate or lageniform, thin-walled, hyaline. *Hymenophoral trama* irregular, consisting of 3.0–7.6 μm-wide hyphae. Pileipellis a cutis composed of hyphae of 3.3–5.3 μm in diam., smooth, cylindrical, hyaline; pigment epiparietal, minutely to strongly encrusting.

Habitat and distribution: This species is bryophilous, growing on the moss *Hyophila involuta* (Hook.) A. Jaeger mixing with soil located on a community balcony in a residential area of Wan’an County, Jiangxi Province of China, which is characterized by a subtropical monsoon climate, and is the only known distribution up to now.

Additional specimens examined: CHINA. Jiangxi Province, Ji’an City, Wan’an County, 26.465265° N 114.798753° E, 65 m alt., on moss, 30 March 2022, Z.H. Cao & W.Q. Guo JX002 (HMAS–L 154704), ZRL20220006 (HMAS 281953); 7 April 2022, Z.H. Cao & W.Q. Guo JX003 (HMAS–L 154706).

Notes: The vegetative thalli of this species are so tiny and few in number that they can very easily be ignored. The coexistence of algal cells in the base of the stipe near to the hairs is a very new finding, because previously, algal cells were only reported in the vegetative thallus of basidiolichens and known as green algae *Coccomyxa* [2,34,35]. The algal cells found in the new species are also unicellular and green, and unfortunately, this algal species has not been identified. However, the possibility that they are moss chloroplasts can be excluded, although moss chloroplasts are also unicellular and green, because chloroplasts are organelles within the cells of moss and need to live in the cytoplasm [55], and so it seems unlikely that the moss chloroplasts would escape from the moss cells and exist separately, trapped in the fungal hyphae. Furthermore, the moss chloroplasts are oval and 2.5–3 × 5–6 μm in size (Figure 3K), different from the algal cells (6–10 μm in diam., Figure 3H). Inconspicuous or absent thalli are common in the basidiolichens of Agaricales, for example in the bryophilous or phycophilous basidiolichen genus *Lichenomphalia*, and the thalli are not obvious in some species such as *L. umbellifera* (L.) Redhead, Lutzoni, Moncalvo & Vilgalys and *L. velutina* (Quél.) Redhead, Lutzoni, Moncalvo & Vilgalys [2,34]. The *Botrydina*-type globular thallus of the new species (Figure 3I) is also similar to *L.*
*meridionalis* (Contu & La Rocca) P.A. Moreau & Courtec. [56], and the algal cells and hyphae are also observed (Figure 3J). However, the new species is distant from *Lichenomphalia* spp., but clusters within the genus *Omphalina*, close to *O. pyxidata* (Bull.) Quél. and *O. chionophila* Lamoure in phylogeny (Figure 2), which also have brown caps, small agarics, and decurrent lamellae [57], but are not known to have a lichenized form.

## 4. Discussion

Phylogenetic and morphological analyses support this new species as a member of *Omphalina* s. stricto. In the phylogenetic tree (Figure 2), *O. chionophile* is sister to *O. licheniformis*. However, in morphology, they can be distinguished by the size of the basidospores—that of *O. chionophile* is 8–10 × 5–6 μm [25,54,58], and that of *O. licheniformis* is 4.9–5.5 × 3.8–4.6 μm. Furthermore *O. chionophile* lacks cheilocystidia. The type species *O. pyxidata* is also phylogenetically close to *O. licheniformis,* sharing similar basidiome features which remain difficult to distinguish in situ. However, they can be distinctly separated under a microscope according to the basidiospore and cheilocystidia, as the basidiospores of *O. pyxidata* (7–8 × 5–6 μm) are larger than those of *O. licheniformis* (4.9–5.5 × 3.8–4.6 μm), and the cheilocystidia of *O. pyxidata* are often branched, while the cheilocystidia of *O. licheniformis* are cylindrical and flexuose, narrowly clavate or lageniform, and not branched [53,54].

The following species differ significantly from *O. licheniformis* (4.9–5.5 × 3.8–4.6 μm) in spore size: *O. rivulicola* (8–10.5 × 5.5–7 μm), *O. mutila* (6.5–10 × 4–6 μm), and *O. subhepatica* (6–8 × 4–5 μm) [54,59]. In terms of the presence or absence of cheilocystidia, *O. licheniformis* possesses cheilocystidia, but none of the following species have cheilocystidia: *O. mutila*, *O. demissa*, *O. subhepatica*, *O. chionophile*, *O. galericolor*, *O. kuehneri*, *O.arctia*, *O. rivulicola* [54,60]. In terms of habitat, *O. licheniformis* is found on mosses, but the following species are found in other habitats: *O. mutila* grows on humid soil in heathland and marshes with *Calluna*, *Erica* and *Mokunia*; *O. chionophile* grows on naked solifluction soil; and *O. pyxidata* and *O. rivulicola* grow on dry sandy soil [54]. In terms of the color of basidiomes, *O. licheniformis* is reddish brown to yellowish brown, but *O. mutila* forms white basidiomes [54,61]. Therefore, *O. licheniformis* is a distinguished species in both molecular composition and morphology.

Lichen-forming fungi are an important component of the kingdom Fungi, making up nearly 20% of the known fungal species, among which over 99% of species belong to Ascomycota [1]. Previous studies showed that more losses than gains of lichenization have occurred during the evolution of Ascomycota, resulting in lichen-forming fungi becoming the ancestors of major lineages of non-lichen-forming fungi in Ascomycota [62]. Compared with ascolichens, basidiolichens are much rarer, and comprise less than 1% of species of the known lichen-forming fungi [1]; however, lichenization in the evolution of Basidiomycota is also very important in related research, and is even treated as one of necessary models to study the evolution of lichens [63].

As early as 1995, Gargas et al. found three independent origins of lichenization in Basidiomycota, i.e., coral *Multiclavula* (Coker) R.H. Petersen as the basal origin, gilled mushroom *Lichenomphalia umbellifera* (syn. *Omphalina umbellifera*), and cyanolichen *Dictyonema pavonium* (Weber & D. Mohr) Parmasto. Basidiolichens are known to consist of far more than three genera nowadays, as mentioned above, including 5 orders, 5 families, 15 genera and 172 species, and six to seven independent lichenization events happened in the Basidiomycota, among which over 85% (147 species) belong to eight genera of Hygrophoraceae Lotsy in Agaricales, viz. *Acantholichen* P.M.Jørg., *Arrhenia* Fr., Cora Fr., *Corella* Vain., *Cyphellostereum* D.A.Reid, *Dictyonema* C.Agardh ex Kunth, *Lichenomphalia*, and *Semiomphalina* Redhead [1]. Except for *Corella* and *Semiomphalina*, all of the other six genera of Hygrophoraceae were included in our phylogenetic analyses (Figure 2), among which the genus *Lichenomphalia* was included in *Omphalina* [2].

The genus *Omphalina* has been taken as a model system to study lichenization since the late 1990s due to its variable nutritional modes [4]; however, after a series of species such as *Omphalina umbellifera*, etc., were transferred to other genera, no lichen-forming species were reported in *Omphalina* s.str., until the finding of *Omphalina licheniformis* in this study. Our study indicates that within the Agaricales, the lichenization process also occurred in the Omphalinaceae of the suborder Tricholomatineae. In the previous reports on basidiolichens, algal cells have never been found in the fruiting body structures [2,34,35,63], but indeed existed in the stipe of *Omphalina licheniformis* (Figure 3). This finding of unusual new species updated our understanding of the delimitation of *Omphalina*, indicating that both non-lichen-forming and lichen-forming fungal species are included simultaneously. Moreover, these results provide new insights and evidence for understanding the significance of lichenization during the evolution of Basidiomycota. Through this study, it should be noted that we need to pay more attention to the Basidiomycota fungi, especially to whether the algal cells are present in the fruiting bodies, which would be very helpful to distinguish more potential basidiolichens and explore the cryptic species diversity through these algal cell examinations.

The presence of algal cells as lichen photobionts is well-known to provide a carbon source for the mycobiont [64], which is relatively easy to understand in ascolichens, because fruiting bodies such as apothecia and pycnidia are closely connected parts of the lichen thallus, and the photobiont can be found both in the thallus and fruiting bodies, except the lecideine-type apothecia and pycnidia without hymenial algae. The algal cells in basidiolichens are assumed to be similar in function [5,65], but there is still an absence of strong evidence, especially due to the fruiting bodies of basidiolichens, which often look separable from the thallus in most cases, and algal cells have only been reported in the thallus previously [2,34,35,63]; moreover, sometimes the thallus is not obvious [2,34].

## Figures and Tables

**Figure 1 jof-08-01033-f001:**
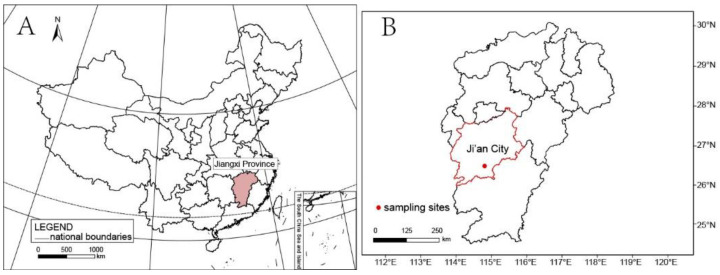
Collection site. (**A**). Jiangxi Province is colored with pink. (**B**). Ji’an City is circled with a red line and the collection site is marked with a red solid circle.

**Figure 2 jof-08-01033-f002:**
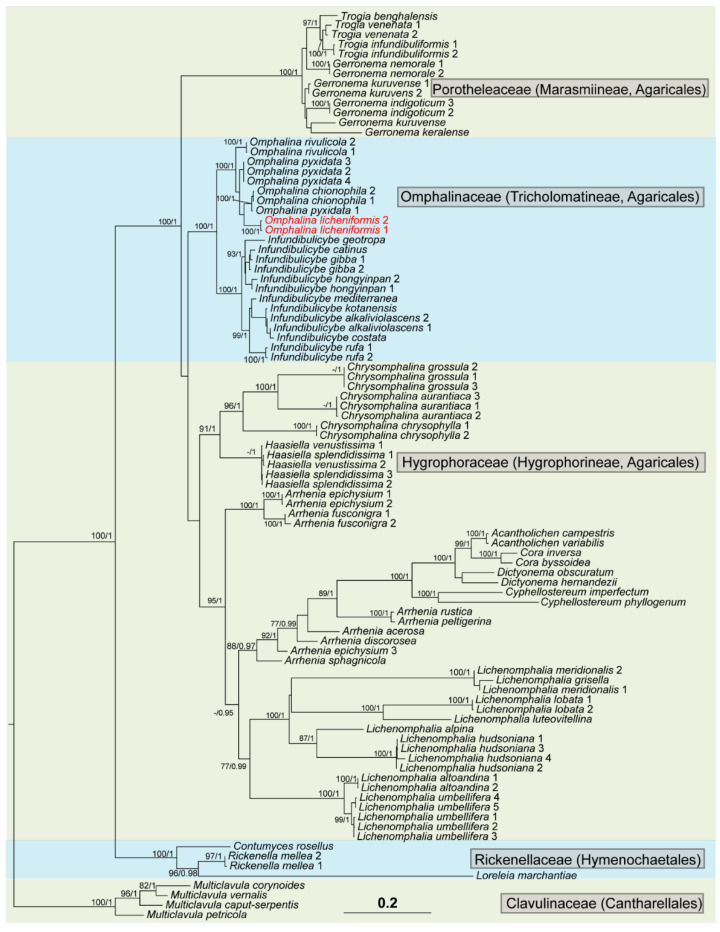
The RAxML tree of omphalinoid species based on the concatenated ITS + nuLSU dataset. The numbers in each node represent bootstrap support (BS) and posterior probability (PP) values. BS values ≥ 75% and PP values ≥ 0.95 were plotted on the branches of the tree. The samples corresponding to the new species are in red. Scale in 0.2 substitution per site.

**Figure 3 jof-08-01033-f003:**
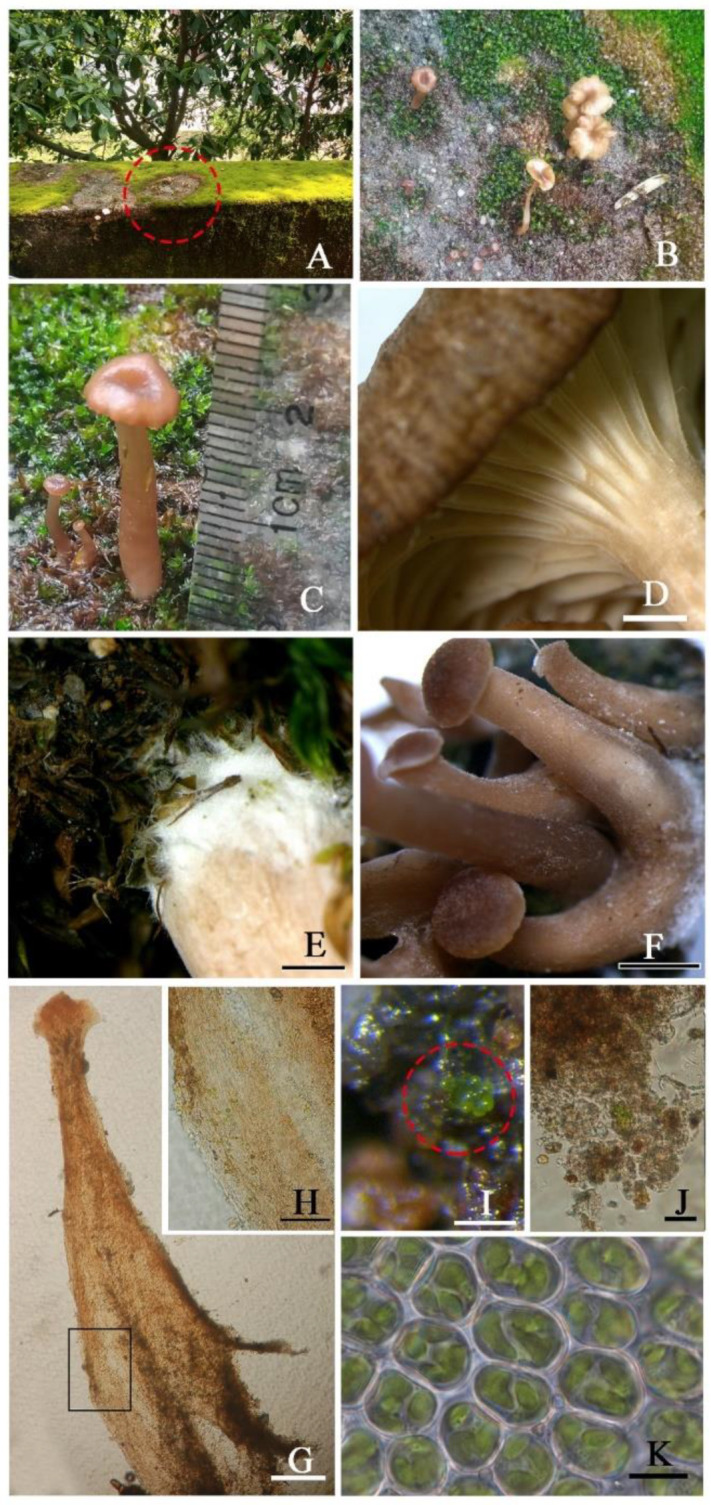
The habitat and habits of *Omphalina licheniformis* sp. nov (holotype HMAS–L 154705). (**A**) Community balcony in a residential area—the habitat of basidiomata is marked by a red circle. (**B**,**C**) The basidiomata in situ. (**D**) Lamellae. (**E**) Hair. (**F**) Young basidiomata. (**G**) Microscopic observation of a young basidiomata with green algae cells in the stipe marked by the black box. (**H**) Zoom in on the black box. (**I**) Tiny and very few *Botrydina*-type vegetative thalli. (**J**) Green algae cells and hyphae in the thallus. (**K**) Leaf cells and chloroplasts inside of the moss *Hyophila involute*. Bars: D-F = 1 mm, G = 200 µm, H-I = 100 µm, J = 20 µm, K = 10 µm.

**Figure 4 jof-08-01033-f004:**
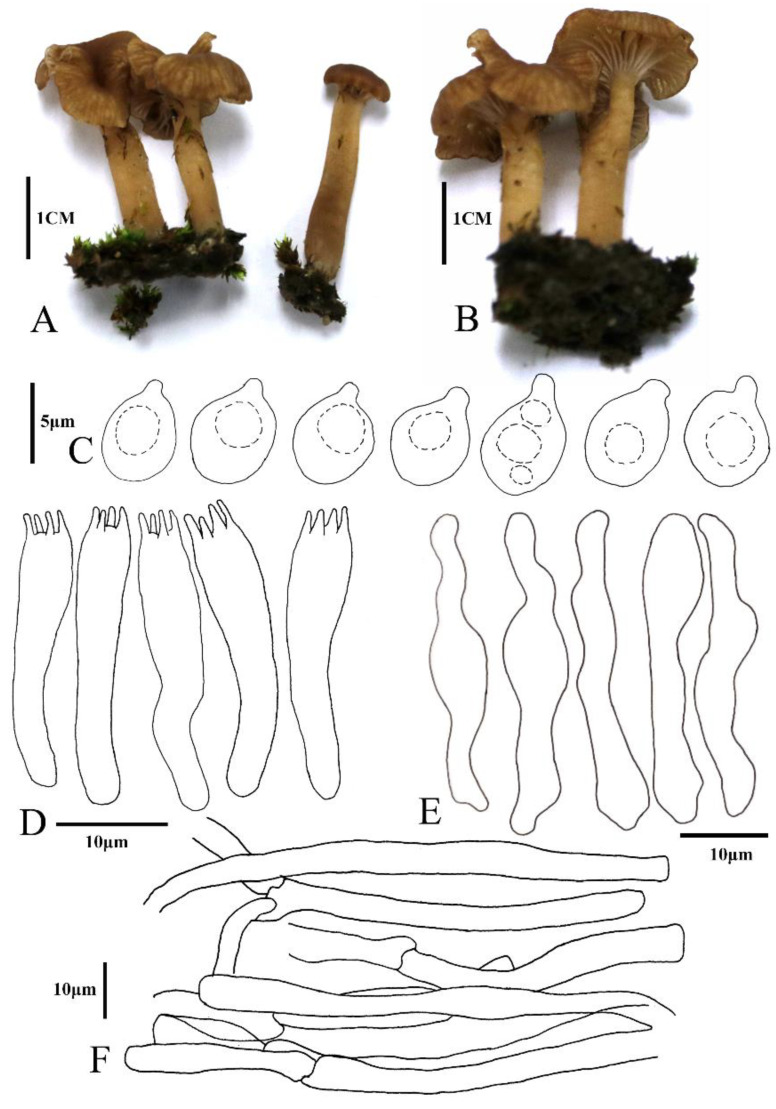
The basidiomes’ habits and the anatomic structure of *Omphalina licheniformis* sp. nov (ZRL 20220005, HMAS 281952, isotype). (**A**,**B**) Basidiomes. (**C**) Basidiospores. (**D**) Basidia. (**E**) Cheilocystidia. (**F**) Pileipellis hyphae.

## Data Availability

Publicly available datasets were analyzed in this study. All newly generated sequences were deposited in GenBank (accessed on 9 June 2022, https://www.ncbi.nlm.nih.gov/genbank/; Appendix A). All new taxa were deposited in Fungal names (accessed on 14 July 2022, https://www.fungalinfo.im.ac.cn).

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
