# Peer review of "New Insights into Lichenization in Agaricomycetes Based on an Unusual New Basidiolichen Species of *Omphalina s.* str."

_jof, 2022, doi:10.3390/jof8101033_

Round 1

Reviewer 1 Report

Interesting paper with sound arguments for the existence of a new species of Omphalina. The lichen habitus of the proposed species is sufficiently demonstrated although it would have been preferred to identify the green algae involved by sequencing. Green algae presence in the stipe of the basidiomycete needs better verification, at least concerning the role these green algae are playing there, are they truly connected to the basidiomycete, is this the same green algae species as can be found in the thallus.

Author Response

Dear reviewer,

Thank you so much for your review and encouraing comment. We added a paragraph at the end of Discussion for explaining the role of green algae, and uploaded a response file for detail. Thanks again.

Xinli Wei

Author Response 

Interesting paper with sound arguments for the existence of a new species of Omphalina. The lichen habitus of the proposed species is sufficiently demonstrated although it would have been preferred to identify the green algae involved by sequencing. Green algae presence in the stipe of the basidiomycete needs better verification, at least concerning the role these green algae are playing there, are they truly connected to the basidiomycete, is this the same green algae species as can be found in the thallus.

Response: Thank you so much for the review and encouraging comments. The presence of green algae in the fruiting bodies is paid attention for the first time in the basidiolichens. Before this, it was only reported in the thallus. About the role of these algae, we understand them to be lichen photobiont. We add a paragraph at the end of Discussion concerning this. The thallus is indeed so tiny and few that it cannot be found easily in Omphalina licheniformis, we have tried to amplify the DNA and culture the algae, but failed, so, except the description of presence of green algae, we have no way to identify them in the current case. We would continuously try in the next summer after we get the fresh specimens.

Reviewer 2 Report

Dear authors,

The manuscript would add a new knowledge regarding lichen-associated Omphalina species. The writing is fine and results from morphological and phylogenetic analyses are done appropriately.  Discussion provides clear statement on distinct characters of the proposed new species. 

I only have a few comments:

1. methods on molecular works and phylogenetic analyses are relatively short. Although the authors cited references that explain details of PCR procedure, ML and BI analyses. Some other information should be added e.g. modeltest and selected best substitution models, how to check congruence among loci, the average standard deviation of split frequency obtained from BI and how to visualize tree. 

2. All texts should be checked again. I can still find some typos. 

Best regards,

Author Response

Dear Reviewer,

Thank you so much for your review and encouraging comment. We carefully checked the text and made the corresponding modifications. Besides, we uploaded a response file for the detailed modifications.

Xinli Wei

Author Response 

The manuscript would add a new knowledge regarding lichen-associated Omphalina species. The writing is fine and results from morphological and phylogenetic analyses are done appropriately. Discussion provides clear statement on distinct characters of the proposed new species. 

Response: Thank you so much for your review and encouraging comment.

I only have a few comments:

  1. methods on molecular works and phylogenetic analyses are relatively short. Although the authors cited references that explain details of PCR procedure, ML and BI analyses. Some other information should be added e.g. modeltest and selected best substitution models, how to check congruence among loci, the average standard deviation of split frequency obtained from BI and how to visualize tree. 

Response: We added the detailed description about the selection of best substitution models and the analyzing process, and added the corresponding references.

  1. All texts should be checked again. I can still find some typos. 

Response: Thanks, we then carefully checked the text and modified correspondingly the typos.